# Evaluation of Epithelial–Mesenchymal Transition Markers in Autoimmune Thyroid Diseases

**DOI:** 10.3390/ijms24043359

**Published:** 2023-02-08

**Authors:** Pablo Sacristán-Gómez, Ana Serrano-Somavilla, Lía Castro-Espadas, Nuria Sánchez de la Blanca Carrero, Miguel Sampedro-Núñez, José Luis Muñoz-De-Nova, Francisca Molina-Jiménez, Alejandra Rosell, Mónica Marazuela, Rebeca Martínez-Hernández

**Affiliations:** 1Department of Endocrinology, Hospital Universitario de la Princesa, Instituto de Investigación Princesa, Universidad Autónoma de Madrid, C/Diego de León 62, 28006 Madrid, Spain; 2Centro de Investigación Biomédica en Red de Enfermedades Raras (CIBERER GCV14/ER/12), 28029 Madrid, Spain; 3Department of General and Digestive Surgery, Hospital Universitario de la Princesa, Instituto de Investigación Princesa, Universidad Autónoma de Madrid, C/Diego de León 62, 28006 Madrid, Spain; 4Gastroenterology Research Unit, Hospital Universitario de la Princesa, Instituto de Investigación Princesa, Universidad Autónoma de Madrid, C/Diego de León 62, 28006 Madrid, Spain; 5Pathology Unit, Hospital Universitario de la Princesa, Instituto de Investigación Princesa, Universidad Autónoma de Madrid, C/Diego de León 62, 28006 Madrid, Spain; 6Faculty of Medicine, Universidad San Pablo CEU, Urbanización Montepríncipe, Alcorcón, 28925 Madrid, Spain

**Keywords:** autoimmune thyroid diseases, Graves’ disease, Hashimoto’s Thyroiditis, TGF-β, EMT, primary cilia

## Abstract

A state of chronic inflammation is common in organs affected by autoimmune disorders, such as autoimmune thyroid diseases (AITD). Epithelial cells, such as thyroid follicular cells (TFCs), can experience a total or partial transition to a mesenchymal phenotype under these conditions. One of the major cytokines involved in this phenomenon is transforming growth factor beta (TGF-β), which, at the initial stages of autoimmune disorders, plays an immunosuppressive role. However, at chronic stages, TGF- β contributes to fibrosis and/or transition to mesenchymal phenotypes. The importance of primary cilia (PC) has grown in recent decades as they have been shown to play a key role in cell signaling and maintaining cell structure and function as mechanoreceptors. Deficiencies of PC can trigger epithelial–mesenchymal transition (EMT) and exacerbate autoimmune diseases. A set of EMT markers (E-cadherin, vimentin, α-SMA, and fibronectin) were evaluated in thyroid tissues from AITD patients and controls through RT-qPCR, immunohistochemistry (IHC), and western blot (WB). We established an in vitro TGF-β–stimulation assay in a human thyroid cell line to assess EMT and PC disruption. EMT markers were evaluated in this model using RT-qPCR and WB, and PC was evaluated with a time-course immunofluorescence assay. We found an increased expression of the mesenchymal markers α-SMA and fibronectin in TFCs in the thyroid glands of AITD patients. Furthermore, E-cadherin expression was maintained in these patients compared to the controls. The TGF-β-stimulation assay showed an increase in EMT markers, including vimentin, α-SMA, and fibronectin in thyroid cells, as well as a disruption of PC. The TFCs from the AITD patients experienced a partial transition to a mesenchymal phenotype, preserving epithelial characteristics associated with a disruption in PC, which might contribute to AITD pathogenesis.

## 1. Introduction

Autoimmune thyroid diseases (AITD) are the most prevalent autoimmune disorders in the global population, with a 5% prevalence, and are most common among middle-aged women [1,2]. They develop as a consequence of tolerance loss against self-thyroid antigens and can be classified into two main types with opposite clinical phenotypes: Hashimoto’s Thyroiditis (HT) and Graves’ disease (GD). In HT, the thyroid gland is seriously damaged by a heavy infiltration of immune cells, which results in thyroid cell apoptosis and hypothyroidism [1]. On the other hand, GD is characterized by follicular hyperplasia and excessive production of thyroid hormones leading to hyperthyroidism. Hyperthyroidism is mainly due to the overactivation of the thyroid-stimulating hormone receptor (TSH-R) by stimulating antibodies (TSH-R-Ab) [1]. An extrathyroidal manifestation commonly observed in GD patients is Graves’ ophthalmopathy (GO). GO is characterized by both an excessive production of glycosaminoglycans or extracellular matrix (ECM) components by orbital fibroblasts (OF) and an immune cell infiltration [3,4]. In these phenotypes, other autoimmune features are present, such as an increase in anti-thyroperoxidase (TPO) and anti-thyroglobulin (TG) antibodies [1,3,5].

After years of research and despite their high prevalence, the molecular mechanisms underlying these diseases are still not completely understood [5]. Several studies have shed light on the genetic predisposition to AITD, reporting relationships of these diseases with several polymorphisms in different genes [6], as well as epigenetic variations [7]. In addition, age, sex, and environmental factors have been related to their development [2,8]. AITD are characterized by a disruption in immune homeostasis, where immune cells such as regulatory T cells (Tregs) show a reduced immune suppression activity, whereas the function of cells with inflammatory phenotypes, such as T helper (Th) 1, Th2, or Th17 cells, is enhanced [5,9,10,11]. These changes, together with the influence of several cytokines, such as interferon gamma (IFN-γ) [12], interleukin 1 beta (IL-1β), tumor necrosis factor alpha (TNF-α), and transforming growth factor beta (TGF-β), create a chronic inflammation environment that can lead to thyroid cell destruction [13,14,15,16,17]. These cytokines can also lead to the acquisition of a mesenchymal phenotype by epithelial cells [18,19,20].

Epithelial–mesenchymal transition (EMT) is a process that occurs under both physiological and pathological conditions, and it is characterized by the loss of epithelial characteristics and the acquisition of mesenchymal features by epithelial cells [21]. Accordingly, this process results in changes in cell behavior, morphology, polarity, cytoskeletal organization, or molecular components in epithelial cells, which lead to cells with increased motility, migration, plasticity, and secretion of ECM components [22]. Briefly, cells lose epithelial markers such as E-cadherin, cytokeratin, or Zonula occludens-1 (ZO-1) and acquire mesenchymal markers such as fibronectin, alpha-smooth muscle actin (α-SMA), fibroblast specific protein-1 (FSP-1/S100A4), or vimentin, among others. Also, mesenchymal cells synthesize ECM components, such as collagen I, which may trigger fibrosis caused by excessive fibrous connective tissue deposition [22,23]. The accumulation of fibrotic components can impair the organ affected. Indeed, EMT is a major feature in several autoimmune diseases, such as rheumatoid arthritis (RA), inflammatory bowel disease (IBD), and primary Sjögren syndrome (pSS) [24,25,26,27,28,29]. In this context, although fibrosis is one of the main pathological characteristics of HT and the transition of orbital fibroblasts to myofibroblasts contributes to the pathogenesis of GO [3], studies on EMT in AITD are scarce [19]. EMT is triggered by several factors, such as genetic/epigenetic alterations, chronic inflammation, and cytokines, such as TGF-β, among others [30,31,32,33,34,35].

By integrating miRNA and mRNA data from AITD thyroid tissue samples, we recently reported dysregulation of primary cilia (PC) as a novel susceptibility pathway that controls AITD pathogenesis. Indeed, the number of PC was dramatically reduced in AITD, and, in some cases, they almost disappeared [36]. PC are defined as individual organelles in a protrusion in the apical surface of the cell. PC trigger several intracellular signal transduction cascades that are indispensable for cell development, proliferation, differentiation, survival, and migration [37]. In thyroid follicular cells (TFCs), PC can also play a role in modulating hormone secretion [38,39]. In addition, it was recently described that the deficiency of primary cilia triggers EMT under resting condition and exacerbates it under the influence of fibrotic signals such as TGF-β [40,41].

Keeping all this in mind and regarding our previous results on primary cilia defects in AITD [36] and their possible involvement in EMT, we evaluated the expression of epithelial and mesenchymal markers in thyroid tissue samples from AITD and correlated their expression with patients’ clinical outcomes. Our data indicate that there is an increase in the acquisition of mesenchymal markers by TFCs in AITD that could contribute to the pathogenesis of these diseases. Furthermore, EMT induction by TGF-β in thyroid cells suggests the potential usefulness of this pathway as a novel therapeutic avenue to treat AITD.

## 2. Results

### 2.1. EMT Markers in AITD

Concomitant expression of epithelial and mesenchymal markers is often used to identify cells that are undergoing EMT. Thus, we first analyzed RNA levels of epithelial and mesenchymal markers within thyroid tissue from AITD patients and controls. E-cadherin (*CDH1*), a marker of epithelial cells, was significantly downregulated in HT tissue in comparison to the control and GD thyroid tissue (mean relative expression 0.05 in HT vs. 0.13 in controls and 0.14 in GD; *p* = 0.001 and 0.0001, respectively) (Figure 1A). Interestingly, when we correlated the expression levels of the different markers with clinical parameters, we observed that *CDH1* expression had a strong inverse correlation with thyrotropin (TSH) (r = −0.7680; *p* < 0.0001) and a significant positive correlation with levels of free-T4 (FT4) (r = 0.7833; *p* = 0.0172) and TSH-R-Ab (r = 0.73 and *p* = 0.045) (Figure 1B).

Regarding markers of mesenchymal cells, vimentin (*VIM*) was significantly upregulated in GD compared to HT tissue (mean relative expression 4.75 vs. 2.53, respectively; *p* = 0.0062); however, no significant differences were found when compared to the controls. Although RNA levels of α-SMA (*ACTA2)* and fibronectin (*FN1)* did not exhibit a significant variation between AITD and control thyroid tissue, *FN1* had a tendency to be upregulated in HT compared to control samples (Figure 1A).

In order to confirm these results and identify cells undergoing EMT, we studied the protein expression of these markers with immunohistochemistry in 50 thyroid samples. Regarding the epithelial marker E-cadherin, we did not observe significant differences between thyroid follicular cells from HT, GD, and control tissues (Figure 2A). Regarding mesenchymal markers, we observed a significant increase in AITD for fibronectin (mean immunohistochemistry [IHC] score of 1.15 in HT and 1.25 in GD vs. 0.1563 in control thyroid tissues; *p* = 0.0001 in both cases) and α-SMA (mean IHC score of 1.68 in HT and 1.80 in GD vs. 0.89 in control thyroid tissues; *p* = 0.0194 and *p* = 0.0070, respectively) (Figure 2B–D). Although we did not detect an increase in vimentin expression, we observed a significant differential distribution pattern in the GD tissue with a location change from a cytoplasmic and perinuclear staining to a peripheral cell distribution toward the basal membrane (basal vimentin mean expression 1.06 in GD vs. 0.22 in control thyroid tissues; *p* < 0.0001) (Figure 2E). Regarding correlations between EMT markers and clinical parameters, positive significant correlations were observed between fibronectin and basal vimentin, fibronectin and α-SMA, and α-SMA and TSH-R-Ab (Figure 2F).

Although an increase in fibronectin and α-SMA was found by immunohistochemistry, this increase was not corroborated by western blot (WB). However, we detected an increase in total vimentin (mean expression 1.51 in HT, 1.10 in GD, and 1.21 in AITD vs. 0.50 in control tissues; *p* = 0.0091, *p* = 0.0279 and *p* = 0.0026, respectively) and cleaved vimentin (mean expression 0.90 in GD and 0.68 in AITD vs. 0.24 in controls; *p* = 0.0135, *p* = 0.0047, respectively). We also evaluated the expression of ADP-ribosylation factors, such as GTPase 13B (Arl13b), which localize to the cilia. We observed a significantly decreased expression of Arl13b in the AITD tissues compared to the controls (0.23 in HT and 0.25 in AITD tissue vs. 1.27 in controls; *p* = 0.0176 and *p* = 0.0227, respectively) (Figure 3).

### 2.2. TGF-β Stimulation of Cultured Thyroid Cells

TGF-β is one of the main agents involved in the acquisition of mesenchymal markers by epithelial cells and in their loss of epithelial characteristics [30,31]. In fact, adding TGF-β to epithelial cells in vitro is a suitable method to induce EMT in different cell models [33,42]. To study the possible role of EMT in AITD using in vitro models, we induced EMT in the thyroid cell line NThy-ORi 3.1 via stimulation with TGF-β for 48 and 72 h, as previously described [24].

In the TGF-β stimulated cells, we observed an upregulation of RNA levels of *FN1* (0.54 in controls vs. 3.65 in TGF-β stimulated cells; *p* = 0.0022) and *ACTA2* genes (0.0038 vs. 0.0099; *p* = 0.0087). Although we did not observe significant changes in *CDH1* and *VIM*, *VIM* expression had a tendency to increase in TGF-β stimulated cells compared to the controls (*p* = 0.0649) (Figure 4A).

Next, we analyzed the protein levels of these markers in cell homogenates using WB. The significant differences observed in RNA were not found in protein levels, which showed a tendency to an increased expression of FN and VIM at 72 h (Figure 4B).

### 2.3. Primary Cilia Disruption after TGF-β Stimulation

Considering the possible role of PC in the pathogenesis of AITD [36] and the association of these structures with EMT [40,41], we performed a morphometric analysis of PC in serum-starved NThy-ORi 3.1 cultured cells at 24, 48, and 72 h after EMT induction by TGF-β. Arl13b antibody was used to identify PC (Figure 5A). As expected, in non-stimulated cells, we observed an increase in cilia length and number in a time-dependent manner. Interestingly, during the TGF-β stimulation, we observed a significantly reduced cilia length (from 2.9 μm to 2.57 μm, *p* = 0.0019) and frequency of cilia (from 35% to 24%, *p* = 0.0266) at 24 h compared to non-stimulated cells. As stimulation time progressed, this reduction was maintained at 48 h (mean length 2.67 μm vs. 2.33 μm, *p* = 0.0002; frequency 41% vs. 20%, *p* = 0.0003) and at 72 h (mean length 3.07 μm vs. 2.5 μm, *p* < 0.0001, and frequency 55% to 24%, *p* < 0.0001) (Figure 5B).

## 3. Discussion

TGF-β plays a pivotal role in normal human immune response and is involved in the pathophysiological spectrum of thyroid autoimmunity [5,13,17,43]. Based on the clear interplay between TGF-β, EMT, and primary ciliogenesis [30,31,33,40,41], we studied EMT markers in AITD and found that thyroid follicular cells can acquire mesenchymal markers and still preserve their epithelial phenotype. Furthermore, the stimulation of human thyroid cell lines with TGF-β upregulated the mesenchymal markers and disrupted primary cilia, suggesting a possible role of this mechanism in the pathogenesis of AITD.

EMT can be classified into three types: developmental (Type I), fibrosis and wound healing (Type II), and pathological (Type III). Type III is usually associated with cancer progression and inflammation [22,23]. In autoimmune disorders, the pro-inflammatory environment affects cells within tissues. If this environment persists, alteration in the wound-healing process can lead to the accumulation of mesenchymal cells, resulting in fibrosis and atrophy of the organ. This scenario leads to organ failure and the spreading of fibrotic cells to nearby healthy areas [44].

E-cadherin is a calcium-dependent tight-junction protein expressed on the cell membrane. Its main function is to maintain cell–cell adhesion, and its loss is one of the main hallmarks of EMT [45,46]. However, in some kinds of tumors, such as pancreatic cancers, tumor cells do not experience a downregulation of E-cadherin expression and yet preserve their epithelial phenotype, also exhibiting motility and the ability to migrate to other tissues [47]. Studies on rheumatoid arthritis (RA), an autoimmune disorder characterized by synovial tissue hyperplasia, have described a widely spread E-cadherin pattern in patients’ synovial tissue that was related to cell hyperplasia. This expression pattern confirmed that synoviocytes had both epithelial and mesenchymal features due to the influence of arthritic synovial fluid [24]. In the context of AITD, in a study performed in HT samples with RET gene rearrangements, the authors observed a decreased expression of *CDH1* in RET^+^ HT patients, suggesting an association between RET activation and the loss of cell adhesion [48]. In our analysis, *CDH1* expression levels were downregulated in HT tissue compared to the controls and GD tissue. Furthermore, *CDH1* expression levels had a direct correlation with FT4 and TSH-R-Ab levels and an inverse correlation with TSH levels. In GD, hypertrophy and hyperfunction of thyroid follicular cells secondary to the presence of TSH-R-Abs lead to increased FT4 levels. Thus, the correlation of CDH1 levels with FT4 and TSH-R-Ab is probably related to the increase in the number and functions of TFCs. On the contrary, the reduction of this marker in HT can be related to the partial loss of TFCs with epithelial phenotype and the increase in fibrosis associated with HT.

Vimentin is a major component of intermediate filaments, and it is widely expressed in mesenchymal cells. It is overexpressed in several epithelial cancers and is recognized as one of the EMT markers. The upregulation of this protein is associated with an increase in focal adhesions, cell motility, and cytoskeletal reorganization [49,50]. Vimentin cleavage by caspases produces a form of the protein that is associated with a disruption in cytoskeletal organization and, at a final stage, with apoptosis [51]. Indeed, a more diffuse cytoplasmic distribution pattern of vimentin with a stronger staining near the basal membrane has been previously described in AITD. This pattern was attributed to a more proliferative state or hyperplasia [52]. Although in later studies these changes were not considered to be indicative of a specific thyroid pathologic condition [53], we found an increase in vimentin expression levels in AITD patients. Furthermore, the increase in cleaved protein could be explained by the apoptosis of TFCs associated with the progression of HT and by the basal distribution pattern observed in GD tissue samples.

α-SMA is a protein commonly expressed in vascular smooth-muscle cells or myofibroblasts with a main role in fibrogenesis [54,55]. α-SMA expression in fibroblasts correlates with their activation state; i.e., α-SMA levels positively correlate with the number of extracellular matrix proteins produced by fibroblasts [56]. In our study, although α-SMA expression assessed by RNA and WB did not change in AITD samples compared to controls, we could observe a clear increase of α-SMA expression in TFCs from AITD through immunohistochemical analysis. These results suggest that some TFCs experience a partial phenotype transition to mesenchymal cells. Indeed, an increased α-SMA expression was reported in thyroid tissue fibroblasts of patients with GD and HT, showing that these cells had many similar features to orbit fibroblasts that differentiate from myofibroblasts [57].

Fibronectin (FN) is an extracellular protein that acts as a scaffold between cells and components of the extracellular matrix [58]. FN is commonly upregulated among other markers in in vitro EMT-induction models [59]. However, the use of this protein as an EMT marker is partially limited as it is produced by many cell types, including not only fibroblasts but also epithelial cells or mononuclear cells [60,61]. In RA, the increased expression of FN was associated with the induction of pro-inflammatory responses and disease progression [62]. In our study, we found a significantly increased staining for FN in AITD TFCs when compared to control tissues. However, as observed with α-SMA, the analysis of bulk tissue using RNA and WB did not show a significant change.

Regarding the markers evaluated by immunohistochemistry, staining was heterogeneous, and some mesenchymal markers were not expressed in the whole tissue, with some areas presenting an increased staining, especially those closer to immune infiltrates or connective tissue. This could be explained by the fact that, in the context of AITD, immune cells secrete cytokines such as IL1-β, TGF-β, or TNF-α that could contribute to the induction of genes related to a mesenchymal phenotype. For example, the synergy between IL-1β and TGF-β could lead to an increase in TGF-β-induced EMT [18]. Regarding AITD, a study performed in a mouse model of granulomatous experimental autoimmune thyroiditis (g-EAT) showed that, at the early stages of the disease, TGF-β induces an immunosuppressive environment. However, at the final stages, this cytokine plays a profibrotic role. Thus, g-EAT thyroid samples with fibrosis showed a higher presence of TGF-β and TNF-α than the control samples [19].

Although we observed the acquisition of these markers through immunohistochemistry, our analysis using WB did not corroborate these results, showing only an increase in cleaved vimentin and a decrease in Arl13b in protein lysates. One of the advantages of immunohistochemistry is the detection of the exact location (namely, specific cells) of a target protein within a tissue sample. On the other hand, WB gathers quantitative information on global protein levels in the tissue. Thus, the differences observed with the different methods could also be related to the effect on WB determinations of a more vascularized tissue or of a higher content of fibrous or connective tissue in AITD, which also express these markers.

Another key point is the dynamic characteristics of EMT, in which cells progressively acquire mesenchymal markers without a concomitant complete loss of epithelial markers. The expression of both mesenchymal and epithelial markers reflects the plasticity of cells depending on their environment [63,64]. Therefore, EMT does not define the final fate of a cell since this process is reversible and there is also a mesenchymal–epithelial transition (MET) where mesenchymal cells can reacquire an epithelial phenotype. In the context of chronic epithelial degradation, only a few cells experiment with a transition to a mesenchymal state [65], immunostaining being the gold standard technique to analyze them, as bulk tissue analysis would not detect these alterations.

Regarding the possible role of PC in the pathogenesis of AITD [36], we also analyzed the effect of TGF-β on these structures. We showed a decrease in the number and length of PC in TGF-β stimulated cells. PC are involved in TGF-β signaling, as the receptors (TGFBRI and TGFBRII) for this cytokine are expressed at the cilia base [66]. In chondrocytic cells, TGF-β was reported to reduce the levels of intraflagellar transport 88 (IFT88), which is expressed in the cilia, leading to a reduction in cilia length and frequency [67]. Furthermore, PC were disrupted in a kidney epithelial cell line undergoing a TGF-β induced EMT, as they changed their morphology to become longer and increased the expression levels of α-SMA and collagen III genes [40]. In light of these results, we can consider that TGF-β may also be involved in PC alteration.

This study has some limitations. First, we analyzed the main commonly used EMT markers to study EMT. Other markers, such as N-cadherin; different types of collagen, i.e., collagen III, fibroblast secreted protein 1 (FSP-1); transcription factors, such as Snail or Twist family; signaling pathways, such as the Sonic hedgehog (Shh) pathway or the Wnt-β-catenin signaling pathway, can be included in future studies. Second, in culture models, we tried to establish an in vitro model with primary TFCs derived from patients; however, since the number of samples was a limiting factor, an alternative based on a human thyroid cell line was chosen.

To conclude, we have reported the acquisition of mesenchymal features by TFCs in patchy areas of the thyroid, which can be attributed to a transition to myofibroblasts expressing mesenchymal markers. TGF-β, a cytokine involved in thyroid autoimmunity, can be related to the acquisition of this phenotype. Finally, primary cilia disruption may represent a potential research area within the study of AITD and the acquisition of mesenchymal phenotypes by TFCs.

## 4. Materials and Methods

### 4.1. Patient Samples

Thyroid tissue samples were collected from surgeries from AITD patients at the Hospital Universitario de la Princesa and from non-thyroid pathology laryngectomy samples or healthy organ donors at the Institut d’Investigació en Ciències de la Salut Germans Trias i Pujol (IGTP-HUGTIP) Biobank. Clinical diagnoses were all reviewed by a single experienced endocrinologist based on standard clinical, laboratory, and histological criteria. Serum free thyroxine (FT4), thyroid-stimulating hormone (TSH), and antibodies against thyroglobulin (TG), thyroperoxidase (TPO), and TSH receptor (TSH-R) were determined in all patients at the time of the surgery. Clinical data are shown in Table 1.

This study was approved by the Internal Ethical Review Committee of Hospital Universitario de la Princesa (Committee Register Number: 2796, approval date: 26 May 2016), and written informed consent was obtained from all patients in accordance with the Declaration of Helsinki.

### 4.2. RNA Isolation and RT-qPCR

RNA from fresh-frozen thyroid tissues (10 HT, 10 GD, and 10 control samples) were isolated with the miRNeasy Mini Kit (Qiagen) according to the manufacturer’s instructions, and the quality and quantity of RNA were evaluated by NanoDrop ND-1000 analysis. First-strand cDNA was generated with a high-capacity cDNA reverse transcription kit with a ribonuclease inhibitor (Applied Biosystems. Waltham, MA, USA), and quantitative reverse transcription–polymerase chain reaction (RT-qPCR) was performed in triplicate using SYBR Green qPCR Master Mix (Thermo Fisher Scientific. Waltham, MA, USA). A list of primers is included in Table 2, and the reaction was performed with the CFX384 Touch Real-Time PCR Detection System (Bio-Rad. Hercules, CA, USA). Ct values were normalized by the Ct of housekeeping genes such as β-actin and GAPDH.

### 4.3. Tissue Microarrays

A total of 49 formalin-fixed, paraffin-embedded (FFPE) tissues were evaluated using tissue microarrays (TMAs). Of these, 30 were AITD thyroid samples with pathological diagnosis of HT and GD (17 and 16, respectively), and 16 corresponded to control thyroid samples from surgeries at the Hospital Universitario de la Princesa. All samples had a duplicate in the same TMA and were taken and managed in accordance with local regulations with the approval of the local institutional review board. Clinical data are shown in Table 3.

### 4.4. Immunohistochemistry

FFPE samples from healthy controls and patients with AITD were collected and processed in order to obtain tissue sections 3 μm in thickness. In the case of TMAs, a preincubation of the slides with Clear Rite at 65 °C for 15 min was performed with the aim of removing the excess of paraffin. Antigen retrieval was performed in an Agilent Dako PTlink (Agilent. Santa Clara, CA, USA) in a basic or acid buffer, depending on the antibody requirements. Endogenous peroxidase was inhibited with a peroxidase-blocking solution (Dako. Santa Clara, CA, USA). Then, tissue sections were incubated overnight at 4 °C with primary antibodies against E-cadherin (Thermo Fisher Scientific. Waltham, MA, USA. Cat# 33-4000, RRID:AB_2533118), Vimentin (Thermo Fisher Scientific. Waltham, MA, USA. Cat# PA5-27231, RRID:AB_2544707), α-SMA (Thermo Fisher Scientific. Waltham, MA, USA. Cat# PA5-18292, RRID: AB_10980764), and Fibronectin (Thermo Fisher Scientific. Waltham, MA, USA. Cat# PA5-29578, RRID:AB_2547054). The following day, sections were incubated with the proper secondary antibodies conjugated to horseradish peroxidase. Finally, tissue sections were incubated with 3,3′-Diaminobenzidine (DAB), counterstained with hematoxylin (Sigma-Aldrich. San Luis, MO, USA), dehydrated in alcohol, cleared with xylene, and mounted.

### 4.5. Immunohistochemistry Score

Immunohistochemistry quantification was determined by analyzing the intensity of staining in the case of α-SMA and fibronectin and assessing the intensity and basal or cytoplasmatic expression for vimentin. The IHC score was graded as follows: for α-SMA, 0 is for negative staining, 1 is for light staining, 2 is for moderate staining, and 3 is for intense staining; for fibronectin, 0 is for negative staining, 1 is for light staining, 2 is for intense staining; and for vimentin, 0 indicates cytoplasmatic expression, 1 is for low basal expression, and 2 is for wide basal expression.

### 4.6. Thyroid Cell Cultures

Cell cultures were performed with the human thyroid cell line NThy-ORi 3-1 (ECACC 90011609, kindly provided by Dr. Pilar Santisteban, Instituto de Investigaciones Biomédicas “Alberto Sols”, Madrid, Spain). The NThy-ORi 3-1 cell line was cultured in RPMI 1640 medium supplemented with Gluta-MAX, 10% fetal bovine serum or FBS (Hyclone. Logan, UT, USA), and 1% of penicillin/streptomycin (Gibco. Carlsbad, CA, USA).

### 4.7. TGF-Β Stimulation Assays

Cells were cultured until reaching confluence. Thereafter, cells were stimulated or not with TGF-β 10 ng/mL (Miltenyi Biotec. Bergisch Gladbach, Germany) in serum-free DMEM. TGF-β was left for 24 h, 48 h, and 72 h. Then, cells were washed and collected in TRiZol for RNA extraction, then scrapped and resuspended in RIPA + Protease and phosphatase inhibitor cocktail Halt^TM^. (ThermoFisher Scientific. Waltham, MA, USA) for WB and in round coverslips for immunofluorescence analysis.

### 4.8. Immunofluorescence Microscopy Analysis

Cells were cultured on round coverslips in 6-well plates, as previously described [7]. Briefly, cells were washed with PBS and fixed with 4% paraformaldehyde. Later, cells were permeabilized with PBS 0.1% Triton X-100 at room temperature and blocked with 5% bovine serum albumin and 10% BSA-PBS.

Cells were incubated with an anti-Arl13b antibody (Proteintech. Rosemont, IL, USA. Cat# 17711-1-AP, RRID:AB_2060867) overnight at 4 °C. Then, slides were incubated for 1 h with an Alexa Fluor 568 labeled goat anti-mouse IgG antibody (Thermo Fisher Scientific. Waltham, MA, USA. Cat# A-11031, RRID: AB_144696). Finally, cell nuclei were counterstained with 4′,6-diamidino-2-phenylindole (DAPI) and analyzed in a Leica Sp5 confocal microscope (Leica Biosystems. Wetzlar, Germany).

The frequency of cilia was estimated manually by analyzing Z-stacked images captured in a confocal microscope. The frequency of ciliated cells was estimated by analyzing the relative number of PC vs. the number of nuclei. A total of 1307 nuclei in non-stimulated cells and 1372 nuclei in stimulated cells were analyzed. The PC length was measured using the ROI measurement tool of ImageJ 1.52i software (National Institutes of Health. Bethesda, MD, USA) for a total of 576 cilia in non-stimulated and 302 in stimulated cells.

### 4.9. Western Blot Analysis

Thyroid tissue samples were mechanically disaggregated in liquid nitrogen and resuspended in RIPA buffer (Sigma-Aldrich. San Luis, MO, USA) containing a protease inhibitor cocktail Halt^TM^ (Thermo Fisher Scientific. Waltham, MA, USA). After 30 min on ice, samples were sonicated, and cell lysates were centrifuged at 4 °C. The supernatant was recovered and stored at −80 °C until use.

Protein samples obtained from the NThy-ORi 3-1 cell line were lysed in RIPA buffer with protease inhibitors at 4 °C in shaking conditions. Later, cells were scrapped and sonicated, followed by a centrifugation step. The resultant supernatant was transferred to another tube and stored at −80 °C until use.

WB was performed in an 8–15% mini-protean TGX precast gel (Bio-Rad. Hercules, CA, USA) and transferred to nitrocellulose membranes. Membranes were blocked and incubated overnight at 4 °C with primary antibodies against E-cadherin (Thermo Fisher Scientific. Waltham, MA, USA. Cat# 33-4000, RRID:AB_2533118), Vimentin (Thermo Fisher Scientific. Waltham, MA, USA. Cat# PA5-27231, RRID:AB_2544707), α-SMA (Thermo Fisher Scientific. Waltham, MA, USA. Cat# PA5-18292, RRID: AB_10980764), fibronectin (Thermo Fisher Scientific. Waltham, MA, USA. Cat# PA5-29578, RRID:AB_2547054), and Arl13b (Proteintech. Rosemont, IL, USA. Cat# 17711-1-AP, RRID:AB_2060867). The next day, membranes were washed with TBS-Tween, incubated with secondary antibodies conjugated to horseradish peroxidase, and visualized using the Pierce^TM^ ECL Western Blotting Substrate chemiluminescent detection reagent kit (Thermo Fisher Scientific. Waltham, MA, USA). After that, membranes were stripped with Restore™ Plus Stripping Buffer (Thermo Fisher Scientific. Waltham, MA, USA) at 37 °C in a shaking incubator, followed by washing steps with TBS-Tween. Finally, membranes were blocked and incubated with anti-β–actin-HRP (Santa Cruz Biotechnology. Dallas, TX, USA. Cat# sc-47778, RRID:AB_626632) polyclonal antibody. ImageJ 1.52i software (National Institutes of Health. Bethesda, MD, USA) was used to quantify the amount of protein in each band.

### 4.10. Statistics

Results were expressed as the arithmetic mean and standard deviation (SD), and differences between groups were compared by the Mann–Whitney or unpaired *t*-test for two-population experiments and one-way ANOVA or Kruskal–Wallis analyses for experiments with more than two populations. Spearman’s rho analyses were performed to detect correlations between the different markers examined by immunohistochemistry and clinical parameters. In addition, *p* values < 0.05 were considered statistically significant. All statistical analyses were performed with the GraphPad Prism 6.0 software (GraphPad Software. Boston, MA, USA).

## 5. Conclusions

In conclusion, an interconnection between EMT, TGF-β, and AITDs is described in this manuscript. TFCs experience a transition or partial transition to mesenchymal cells as they acquire mesenchymal markers, such as fibronectin and α-SMA. This effect is probably caused by the pro-inflammatory microenvironment in AITD and mainly by the influence of TGF-β. TFCs do not lose their epithelial phenotype completely. Finally, PC disruption by TGF-β can contribute to the acquisition of mesenchymal markers by TFCs.

## Figures and Tables

**Figure 1 ijms-24-03359-f001:**
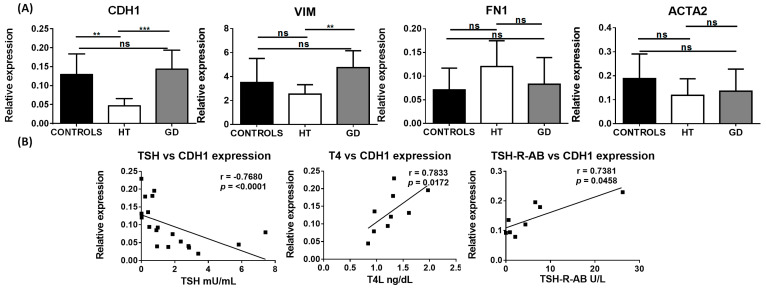
RNA expression of EMT-associated markers in thyroid tissue from controls, HT, and GD patients. (**A**) RT-qPCR expression of E-cadherin (*CDH1*), vimentin (*VIM*), fibronectin (*FN1*), and α-SMA (*ACTA2*) in bulk thyroid tissue samples. Data correspond to the arithmetic mean ± SD. (**B**) Significant correlation analysis of the genes analyzed by RT-qPCR with different clinical laboratory parameters. Abbreviations—ns: not significant, TSH: thyroid-stimulating hormone; FT4: free-T4 hormone; TSH-R-Ab: TSH receptor antibody. ** *p* < 0.01; *** *p* < 0.005.

**Figure 2 ijms-24-03359-f002:**
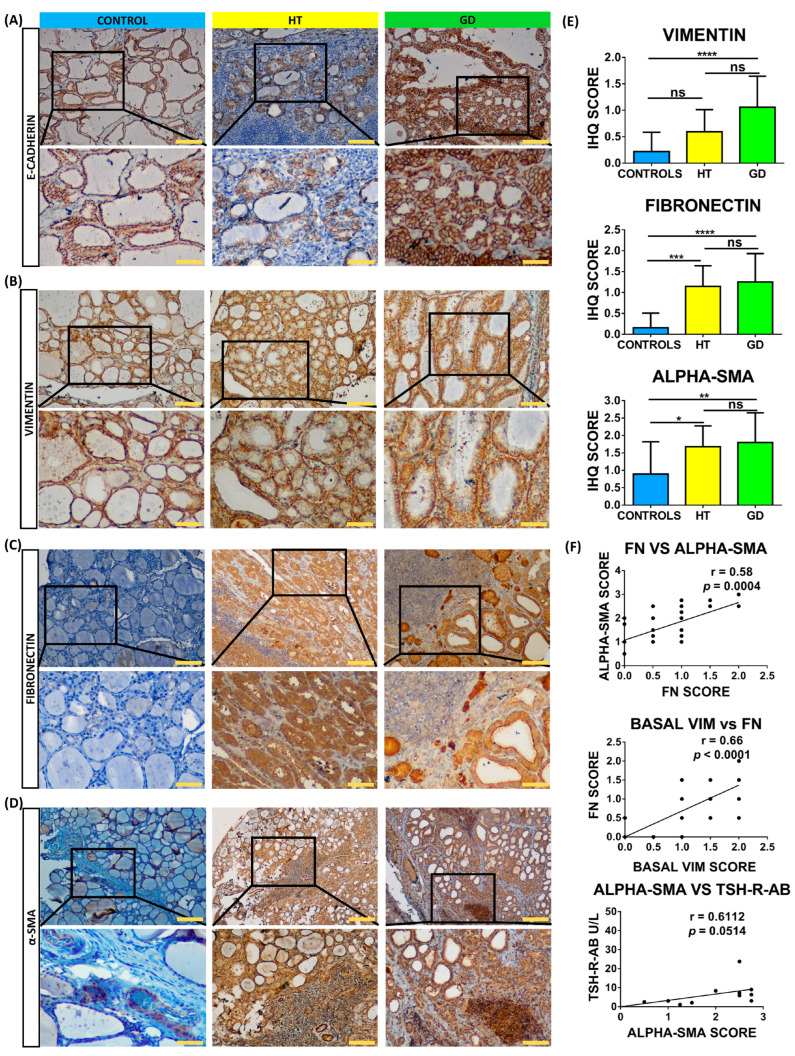
Immunostaining analysis of EMT markers in thyroid tissue from controls, HT, and GD patients. (**A**–**D**) Immunohistochemistry analysis of E-cadherin, vimentin (VIM), fibronectin (FN), and α-SMA. Scale bar A, B, and C: 200 μm, zoom 100 μm. Scale Bar D: 500 μm, zoom 100 μm. (**E**) Immunohistochemistry (IHC) score quantitation of basal vimentin, fibronectin, and α-SMA. (**F**) Correlation of marker IHC score with clinical laboratory parameters. Data correspond to the arithmetic mean ± SD. Abbreviations—ns: not significant. * *p* < 0.05; ** *p* < 0.01; *** *p* < 0.005; **** *p* < 0.0001.

**Figure 3 ijms-24-03359-f003:**
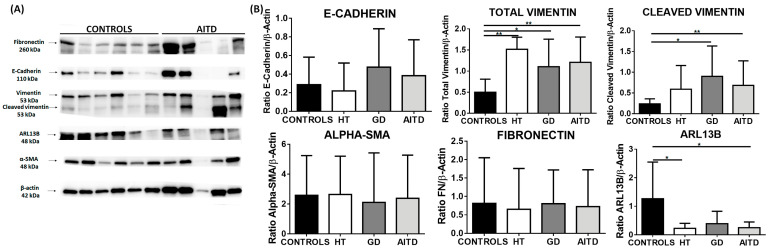
Western blot analysis of EMT markers in thyroid tissue from controls, HT, and GD patients. (**A**) Western blot analysis of E-cadherin, vimentin, fibronectin, α-SMA, and Arl13b. (**B**) Protein quantitation of the EMT markers indicated above. Data correspond to the arithmetic mean ± SD. * *p* < 0.05; ** *p* < 0.01.

**Figure 4 ijms-24-03359-f004:**
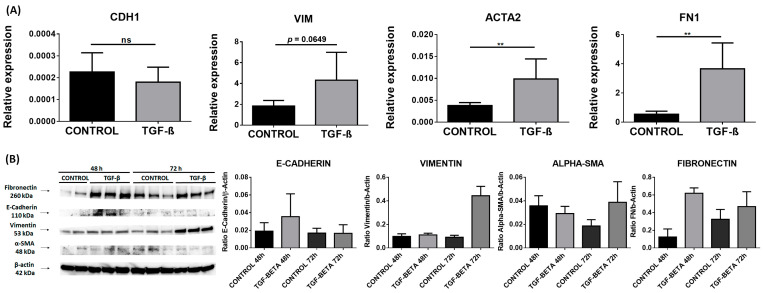
EMT markers in the NThy-ORi 3.1 cell line stimulated with TGF-β. (**A**) RT-qPCR expression of E-cadherin (*CDH1*), vimentin (*VIM*), fibronectin (*FN*), and α-SMA (*ACTA2*). Data correspond to the arithmetic mean ± SD. (**B**) Western blot analysis of E-cadherin, vimentin, fibronectin, α-SMA, and Arl13b. Quantitation is indicated below. Abbreviations—ns: not significant. ** *p* < 0.01.

**Figure 5 ijms-24-03359-f005:**
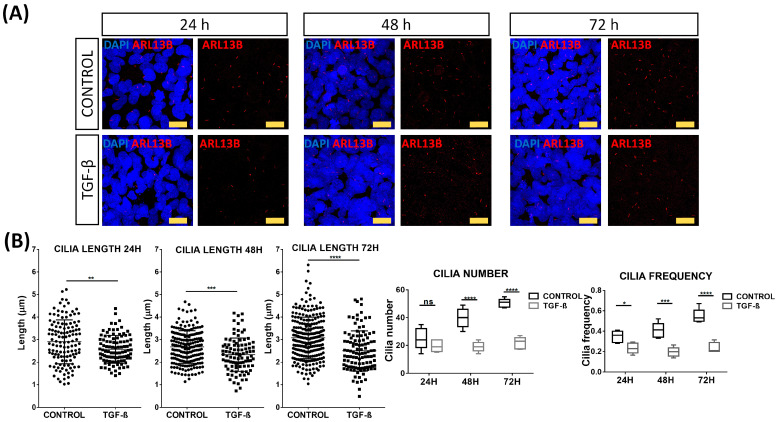
Disruption of primary cilia by TGF-β stimulation. (**A**) Immunofluorescence microscopy analysis of Arl13b (red) expression in absence or presence of TGF-β (10 ng/mL). Cell nuclei are stained with DAPI (blue). Scale bar: 25 μm. (**B**) Quantitation of length, number, and frequency of cilia. Data correspond to the arithmetic mean ± SD. Abbreviations: ns: not significant. * *p* < 0.05; ** *p* < 0.01; *** *p* < 0.005; **** *p* < 0.0001.

**Table 1 ijms-24-03359-t001:** Clinical parameters of AITD patients included in RT-qPCR analyses.

Parameters	HT	GD
N	10	10
Gender (F/M)	10/0	9/1
Age, years	62 (57–70)	47 (40–57)
Ophthalmopathy	0	7
TSH, mU/mL	2.59 (1.68–3.26)	0.44 (0.01–0.8)
T4, ng/dL	-	1.29 (1.02–1.54)
TG-Ab, UI/mL	626 (143.5–728.5)	20 (20–2279)
TPO-Ab, UI/mL	713.5 (434.75–1421.25)	169 (20–578.5)
TSH-R-Ab, U/L	-	5.63 (0.91–7.67)

Values are categorical values and median (interquartile intervals 25–75) for continuous variables. Abbreviations: F, female; M, male T4, thyroxine (normal range = 0.93–1.7); TG-Ab, anti-thyroglobulin antibody (negative < 344); TPO-Ab, anti-thyroid peroxidase antibody (negative < 100); TSH, thyrotropin (normal range = 0.27–4.20); TSH-R-Ab, anti-thyrotropin receptor antibody (negative < 0.7).

**Table 2 ijms-24-03359-t002:** List of primers used in RT-qPCR analyses.

Primer	Orientation	Sequence
CDH1	FORWARD	GCCGAGAGCTACACGTTCAC
	REVERSE	ACTTTGAATCGGGTGTCGAG
VIM	FORWARD	CTCCCTCTGGTTGATACCCAC
	REVERSE	GGTCATCGTGATGCTGAGAAG
FN1	FORWARD	CCTCAATTGTTGTTCGCTGGAGCA
	REVERSE	GGTGACGGAGTTTGCAGTTTC
ACTA2	FORWARD	TGGCTATCCAGGCGGTGCTGTCT
	REVERSE	ATGGCATGGGGCAAGGCATAGC
GAPDH	FORWARD	GCCCAATACGACCAAATCC
	REVERSE	AGCCACATCGCTCAGACAC
β-ACTIN	FORWARD	GCCGACAGGATGCAGAAGGA
	REVERSE	CGGAGTACTTGCGCTCAGGA

**Table 3 ijms-24-03359-t003:** Clinical parameters of patients included in TMA.

Parameters	HT	GD	Controls
N	17	16	16
Gender (F/M)	12/5	15/1	10/6
Age, years	62 (42–66)	48 (43–59)	57 (43–61)
Ophthalmopathy	0	8	0
Smoking	1	6	2
TSH, mU/mL	3.48 (2.15–4.70)	0.15 (0.01–5.04)	1.63 (1.32–2.95)
T4, ng/dL	1.34 (1.11–1.45)	1.05 (0.95–1.51)	1.34 (1.06–1.81)
TG-Ab, UI/mL	80 (20.75–216.25)	95 (23–117)	12 (12–12)
TPO-Ab, UI/mL	86 (22.5–297)	175 (16–223)	4 (4–10)
TSH-R-Ab, U/L	-	5.22 (2.98–8.11)	-

Values are categorical values and median (interquartile intervals 25–75) for continuous variables. Abbreviations: F, female; M, male; T4, thyroxine (normal range = 0.93–1.7); TG-Ab, anti-thyroglobulin antibody (negative < 344); TPO-Ab, anti-thyroid peroxidase antibody (negative < 100); TSH, thyrotropin (normal range = 0.27–4.20); TSH-R-Ab, anti-thyrotropin receptor antibody (negative < 0.7).

## Data Availability

Some or all datasets generated and/or analyzed during the current study are not publicly available but are available from the corresponding author upon reasonable request.

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
