# Peer review of "Evaluation of Epithelial–Mesenchymal Transition Markers in Autoimmune Thyroid Diseases"

_ijms, 2023, doi:10.3390/ijms24043359_

Round 1

Reviewer 1 Report

The manuscript is well written, the ideas are clear and the conclusions are consistent with the aims and the litterature.

The authors made an excellent work.

Simple changes would be, improving the quality of the figures by enlarging them.

Otherwise I suggest the publication of this manuscript.

Author Response

Reviewer 1

The manuscript is well written, the ideas are clear and the conclusions are consistent with the aims and the litterature.

The authors made an excellent work.

Simple changes would be, improving the quality of the figures by enlarging them.

Otherwise I suggest the publication of this manuscript.

Response: We thank the reviewer for the kind suggestions. We have enlarged images and graphs in order to improve the quality of the Figures.

Reviewer 2 Report

See attached file

Author Response

Reviewer 2

Review and comments for the manuscript entitled: Study of epithelial-mesenchymal transition markers in Auto-2 immune Thyroid Diseases.

In the title:

It is suggested that the title shows the type of study performed, as far as possible. This helps to identify the field of study and application of the research.

For example, in this case, it could be convenient to integrate the word "in vitro", and part of the objective of the study which was "To evaluate".

Evaluation of epithelial-mesenchymal transition markers in Auto-2 immune Thyroid Diseases: an invitro study.

In the Abstract:

No suggested changes

In the Keywords:

No suggested changes

In the Introduction:

In lines 43-56: Broad statements and paragraphs are presented without specific bibliographic citations. References are shown without reference. At the end there are 3 references that are 1) very previous and 2) few for the whole text behind them.

It is suggested to update bibliography and support each assertion with scientific literature.

In lines 57-58: same previous suggestions.

In lines 70-78: same previous suggestions.

In general, the paper is of great interest to the field of molecular sciences and specifically within the physiopathology of the thyroid gland.

It is suggested to pay attention to the references not updated and used in the sections of the document, that is, to update them or to leave only those that are of concrete interest and relevance for the manuscript.

Some minimal suggestions were made, which do not minimize, in any way, the hard and important work that the authors have done.

Kind regards

Response: Thank you for your interest in our work and the comments to improve the manuscript.

In the title:

We have changed the title to: “Evaluation of epithelial-mesenchymal transition markers in Autoimmune Thyroid Diseases”

In the Introduction:

We have updated references in the revised version of the manuscript.

Reviewer 3 Report

The paper has the title “Study of epithelial-mesenchymal transition markers in Auto-2 immune Thyroid Diseases”.

It is an experimental study with determination of markers in Thyroid  tissue of persons with autoimmune thyroid diseases (AITD) in comparison with healthy people.

The study is methodical and the authors evaluated the expression of epithelial and mesenchymal markers in thyroid tissue samples from AITD and correlated their expression with patient’s clinical outcomes

They used samples from 20 patients and 10 controls.

They manage to simplify a rather difficult subject and to express it in an understandable way.

The methods and the results are good explain.

The only comments that I have is to mention in their text the reason why all the patient sample, except one, are coming from female patients. This could be of the prevalence of autoimmune thyroid diseases but it could be good the be explain.

In general I do not have further suggestion and I think the paper is worth it to be published.

Author Response

Reviewer 3

The paper has the title “Study of epithelial-mesenchymal transition markers in Auto-2 immune Thyroid Diseases”.

It is an experimental study with determination of markers in Thyroid  tissue of persons with autoimmune thyroid diseases (AITD) in comparison with healthy people.

The study is methodical and the authors evaluated the expression of epithelial and mesenchymal markers in thyroid tissue samples from AITD and correlated their expression with patient’s clinical outcomes

They used samples from 20 patients and 10 controls.

They manage to simplify a rather difficult subject and to express it in an understandable way.

The methods and the results are good explain.

The only comments that I have is to mention in their text the reason why all the patient sample, except one, are coming from female patients. This could be of the prevalence of autoimmune thyroid diseases but it could be good the be explain.

In general I do not have further suggestion and I think the paper is worth it to be published.

Response: We thank the reviewer for the suggestions. We have included in the introduction section the higher prevalence of AITD in women.